# Pregnancy outcomes of women whom spouse fathered children after tyrosine kinase inhibitor therapy for chronic myeloid leukemia: A systematic review

Zsolt Szakács[1,2☯], Péter Jenő Hegyi[1☯], Nelli Farkas[3], Péter Hegyi[1], Márta Balaskó[1], Adrienn Erős[1,2], Szabina Szujó[4], Judit Pammer[4], Bernadett Mosdósi[5], Mária Simon[6], Arnold Nagy[5], Gabriella Für[7], Alizadeh Hussain[4]*

1 Institute for Translational Medicine, Medical School, University of Pécs, Pécs, Hungary, 2 Szentágothai Research Centre, University of Pécs, Pécs, Hungary, 3 Institute of Bioanalysis, Medical School, University of Pécs, Pécs, Hungary, 4 Division of Hematology, First Department of Medicine, Medical School, University of Pécs, Pécs, Hungary, 5 Department of Paediatrics, Medical School, University of Pécs, Pécs, Hungary, 6 Department of Psychiatry and Psychotherapy, Medical School, University of Pécs, Pécs, Hungary, 7 Institute of Pathophysiology, University of Szeged, Pécs, Hungary

☯ These authors contributed equally to this work.
* alizadeh.hussain@pte.hu

**Data Availability Statement:** All relevant data are within the manuscript.

## Abstract

### Introduction

The introduction of tyrosine kinase inhibitors (TKIs) has revolutionized the therapy of chronic myeloid leukemia (CML). Although the efficacy of TKIs is beyond dispute, conception-related safety issues are still waiting to be explored, particularly in males. This systematic review aimed to summarize all available evidence on pregnancy outcomes of female spouses of male CML patients who fathered children after TKI treatment for CML.

### Methods

We performed a systematic search in seven electronic databases for studies that reported on male CML patients who did or did not discontinue TKI treatment before conceiving, and the pregnancy outcomes of their female spouse are available. The search centered on the TKI era (from 2001 onward) without any other language or study design restrictions.

### Results

Out of a total of 38 potentially eligible papers, 27 non-overlapping study cohorts were analyzed. All were descriptive studies (case or case series studies). Altogether, 428 pregnancies from 374 fathers conceived without treatment discontinuation, 400 of which (93.5%) ended up in a live birth. A total of ten offspring with a malformation (2.5%) were reported: six with imatinib (of 313 live births, 1.9%), two with nilotinib (of 26 live births, 7.7%), one with dasatinib (of 43 live births, 2.3%), and none with bosutinib (of 12 live births). Data on CML

**Funding:** Study costs are covered by the Economic Development and Innovation Operative Programme Grant (GINOP 2.3.2-15-2016-000048 to PH, no website available), by Human Resources Development Operational Programme Grants (EFOP-3.6.2-16-2017-0006 to PH, no website available), and by the New National Excellence Programme, Ministry of Human Capacities (ÚNKP-19-3-I to ZS, no website available). The funders had no role in study design, data collection and analysis, decision to publish, or preparation of the manuscript.

**Competing interests:** The authors have declared that no competing interests exist.

status were scarcely reported. Only nine pregnancies (from nine males) and no malformation were reported in males who discontinued TKI treatment before conception.

## Conclusion

Malformations affected, on average 2.5% of live births from fathers who did not discontinue TKI treatment before conception, which is comparable with the rate of malformations in the general population. Large-scale studies with representative samples are awaited to confirm our results.

## Introduction

Chronic myeloid leukemia (CML) is a myeloproliferative neoplasm driven by the presence of the BCR-ABL1 fusion product generated as a result of the *t*(9;22) Philadelphia chromosome (Ph). The annual incidence ranges between 0.4 and 1.75 per 100 000 inhabitants. Although CML can strike at any age and its incidence prominently increases with aging [1], peak incidence falls at around 60 years in Europe [2] but at a lower age in Asia [3].

The introduction of imatinib (IMA), a tyrosine kinase inhibitor (TKI), has revolutionized the treatment of CML, dramatically improving life-expectancies and resulting in a great 10-year survival rate exceeding 80% [4–7]. The success of IMA led to the development of second- and third-generation TKIs, such as nilotinib (NIL), dasatinib (DAS), bosutinib (BOS), and ponatinib (PON). Network meta-analyses confirmed the efficacy and safety of new-generation TKIs in IMA-resistant or IMA-failure cases and even as first-line alternatives of IMA [8, 9].

As a result of the widespread use of TKIs and the subsequent improvement in life quality, hematologists faced new challenges of procreation. Around one-fourth of CML patients, both males and females, are diagnosed at a reproductive age. While women are often in the spotlight, research on male fertility issues is less popular [10–12].

Fertility issues may derive from the molecular mechanism of the agents. TKIs are competitive inhibitors of ABL kinase, inhibiting the autophosphorylation of BCR-ABL, which results in the induction of apoptosis in the corresponding cells. However, TKIs are not purely selective to ABL kinase: targets include c-kit, PDGFR-alpha, c-FMS, and other kinases [13]. This non-selective enzymatic inhibition may interfere with the steps of spermato- and spermiogenesis: sporadic reports indicated that IMA affects the human reproductive system [14, 15]. The most comprehensive report included the semen samples of 48 IMA-treated CML males and proved that IMA is secreted to the semen, reduces sperm survival and activity, but does not significantly affect the levels of gonadotrophic hormones and sexual steroids [16].

Turning to the conception outcomes, the first report that discussed males being exposed to IMA at the time of conception was released in 2003 by Hensley and Ford [17]. Several relevant cases have been reported since then, and expert reviews summarized the available evidence on fertility-related safety issues [10–12, 16, 18–22]. These were all high-quality but non-systematic summaries except in a 2016 review of about 200 cases with a restricted search to one database and other non-electronic data sources [11]. The niche of analyzing disease status at conception has remained unoccupied.

In this study, we aimed to perform a strict systematic review with a transparent, reproducible methodology to summarize conception-related outcomes of TKI-treated males, with a special focus on CML status at conception.

## Methods

This work is reported following the Preferred Reporting Items for Systematic Review (PRISMA) Statement [23]. The pre-protocol of the systematic review was registered *a priori* in PROSPERO under registration number CRD42018087127.

### Search

We performed a comprehensive search of the medical literature. The search strategy covered the following sources:

1. Electronic databases including MEDLINE (via PubMed), EMBASE, Web of Science, Scopus, WHO Global Health Library, Cochrane Controlled Register of Trials (CENTRAL), and ClinicalTrials.gov were searched for relevant reports from 2001 (date of approval of IMA in the US) up to Nov 2020 without other restrictions. We used Medical Subject Heading (MeSH) in combination with free-text terms to capture all relevant papers. The query was designed to include the Chemical Abstract Service (CAS)-numbers of agents: *(chronic AND (myeloid OR myelogenous) AND (leukemia OR leukaemia)) AND ("tyrosine kinase inhibitor\*" OR imatinib OR "152459-95-5" OR nilotinib OR "641571-10-0" OR dasatinib OR "302962-49-8" OR bosutinib OR "380843-75-4" OR ponatinib OR "943319-70-8") AND (pregnant\* OR gestation OR conception OR fertile\* OR inseminate\* OR childbearing OR embryotoxic\* OR genotoxic\* OR teratogenic\*).*

2. Reference lists of relevant included and excluded reports, including previous non-systematic reviews, were hand searched.

3. Citing papers of relevant articles were identified by using Google Scholar.

4. Abstract books of The European Hematology Association (EHA) and The American Society of Hematology (ASH) were hand-searched from 2001 on.

### Selection, eligibility, and data collection

We included records reporting on male patients suffering from CML and receiving TKIs (IMA, NIL, DAS, BOS, or PON) before or at the time of conception if pregnancy-related or neonatological outcomes are available. We excluded cases with cryopreservation of sperm donated before the initiation of TKI treatment. Since unfavorable pregnancy outcomes are rare events, any record containing original data (full-text articles and conference papers) of at least one patient was eligible for inclusion.

All records were combined in a reference manager software (EndNote X7.4, Clarivate Analytics, Philadelphia, PA, US) to remove database overlaps and duplicate references. Then, records were tested against our eligibility criteria by title, abstract, and full-text. Eligible records were subjected to data collection. Two review authors selected the records and collected data in duplicates; discrepancies were resolved by third-party arbitration after each step of selection and data collection. We collected data on patients' baseline characteristics, therapy regimens (agent, dose, timing), pregnancy course; obstetric, neonatal, and pediatric complications; and disease activity of males at conception. We had no contact with the authors of the included papers.

Finishing data collection, we reviewed all records carefully to identify overlaps across study populations. Overlapping records (and data) were linked together, then handled as a cohort of patients.

## Quality assessment

Two review authors used Murad et al.'s tool to assess case studies' and case series' quality in duplicate, resolving discrepancies by consensus [24]. The tool's leading explanatory questions cover four domains: selection, ascertainment, causality, and reporting. As recommended by Murad et al., we did not aggregate scores but discussed the findings as limitations of the evidence.

# Results

## Search and selection

Fig 1 shows the flowchart of the systematic review. A total of 1 957 records were identified in seven databases. Finally, 40 publications reported the pregnancy outcomes of the spouses of male patients. The most common cause of exclusion on full-text assessment was reporting pregnancy outcomes of female CML patients exclusively (57 records).

Out of the 40 publications, we excluded two papers [17, 25]. In the conference paper of Siddique et al. [25], we were unable to separate pregnancy outcomes of IMA-treated males from that of females in a cohort of patients conceiving a total of ten times (the outcomes included three elective terminations; no malformations were recorded). The reason for exclusion was similar in the case of the study by Hensley and Ford [17]: data of IMA-treated CML and gastrointestinal stromal tumor cases were not separable (the outcomes included two elective terminations and two spontaneous abortions; no malformations were recorded).

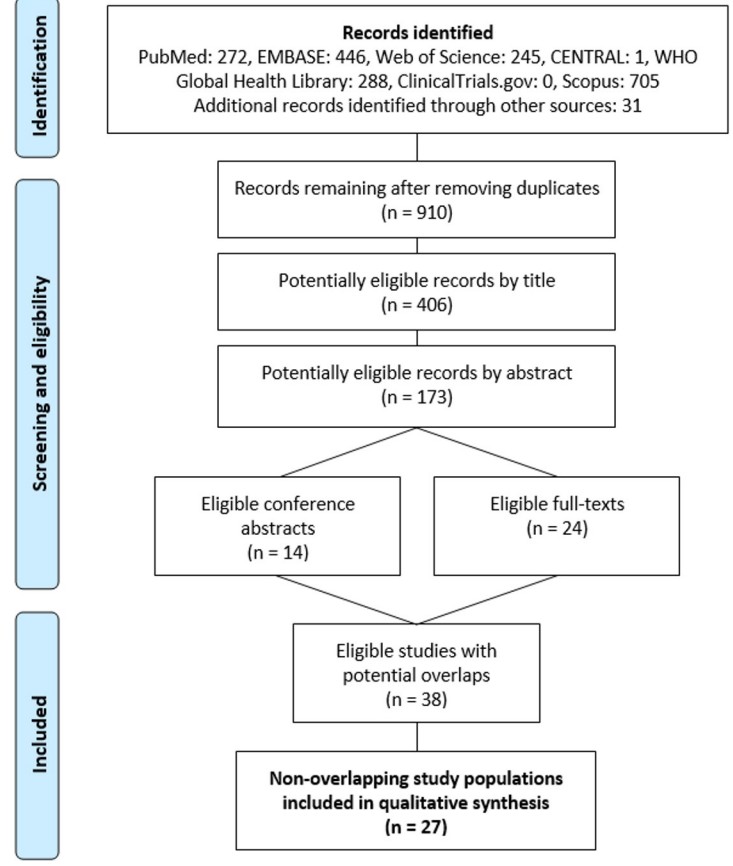

**Fig 1. Flowchart.**

We carefully checked the remaining 38 publications (24 full-text papers and 14 conference abstracts) to find overlaps of cases; finally, 27 non-overlapping cohorts of patients or case studies were identified.

## Characteristics of the studies included

Tables 1 and 2 show the summary of the studies included [10, 11, 26–61]. Seven papers were non-English language articles: one was written in Bulgarian [40], one in French [41], one in Japanese [61], and another four in Chinese [43, 53, 58, 59]. We did not identify any comparative (controlled) studies: all evidence came from descriptive studies (case studies or case series studies). Eleven cohorts of patients were recruited from Europe, another eleven from Asia, two from the US, one from Africa, and there were two multinational studies.

## Planned treatment discontinuation

Nine pregnancies from nine males in three cohorts of patients were reported (Table 1) [10, 11, 26, 38, 41, 49]. Six cases were pre-treated with IMA, another three with DAS. One pregnancy, where the father was pre-treated with IMA, ended up in spontaneous abortion; otherwise, all were uneventful (no malformations were recorded). Detailed follow-up data were not available for the cases. No information is available on NIL, DAS, or BOS. We were unable to separate the outcomes of planned treatment discontinuation (four of 49 males) from those of no treatment discontinuation in one cohort of patients [59].

## No treatment discontinuation

Studies observed a total of 374 males who had conceived under the effect of TKIs (Table 2). A total of 428 pregnancies were reported, 400 of which (93.5%) ended up in a successful delivery

**Table 1. Characteristics of the studies reporting on male patients with planned treatment discontinuation before conception.**

| Study population | Country | N⁰ of pregnancies (N⁰ of males) | TKI (N⁰ of pregnancies) | Non-fatal malformations with live births | *In utero* fatal events (N⁰ of cases, TKI) | Peripartum feto-maternal complications | Infant complications | CML status at conception (N⁰ of cases) | Timing of treatment discontinuation |
|---|---|---|---|---|---|---|---|---|---|
| Abruzzese et al. 2014 [10, 26] (article and conference abstract) and 2016 [11] (article) (from the GINEMA registry) | Italy | 2 (probably 2) | dasatinib (2) | none | none (probably) | none | none | CP (all) | 3 and 5 months before conception |
| Guerci-Bresler et al. 2011 (from the FI-LMC Group) (article in French) [41] | France | 1 (1) | dasatinib (1) | none | none | none | not reported | not reported | 15 days before conception |
| Mukhopadhyay et al. 2015 [49] (article) and Dasgupta et al. 2013 [38] (conference abstract) | India | 6 (6) | imatinib (6) | none | spontaneous abortion (1, imatinib) | not reported | not reported | CP (all), CHR (all), CCR (all), MMR (all) | 4–6 weeks before conception |

CCR, complete cytogenic remission; CHR, complete hematological remission; CML, chronic myeloid leukemia; CP, chronic phase; MMR, major molecular remission; TKI, tyrosine kinase inhibitor

**Table 2. Characteristics of the studies reporting on male patients not discontinuing tyrosine kinase treatment before conception.**

| Study population | Country | N⁰ of pregnancies (N⁰ of males) | TKI (N⁰ of pregnancies) | Non-fatal malformations with live births (N⁰ of cases, TKI) | *In utero* fatal events (N⁰ of cases, TKI) | Peripartum feto-maternal complications (N⁰ of cases, TKI) | Infant complications (N⁰ of cases, TKI) |
|---|---|---|---|---|---|---|---|
| Abruzzese et al. 2014 [10, 26] (article and conference abstract) and 2016 [11] (article) (from the GINEMA registry) | Italy | 44 (probably 40) | imatinib (34), nilotinib (7), dasatinib (1), bosutinib (2) | congenital hip dysplasia (1, imatinib) | none (probably) | premature delivery (1, imatinib) | jaundice (1, imatinib)[1] |
| Alizadeh et al. 2015 [27] (article) | Hungary | 10 (5) | imatinib (8), nilotinib (2) | none | none | none | not reported |
| Aota et al. 2020 [61] (article in Japanese) | Japan | 1 (1) | nilotinib (1) | none | none | none | not reported |
| Assi et al. 2017 [28] (conference abstract) | The US | 7 (7) | nilotinib (5), dasatinib (2) | none | none | not reported | not reported |
| Ault et al. 2006 [29] (article) | The US | 9 (8) | imatinib (9) | gut malrotation (1, imatinib) | spontaneous abortion (1, imatinib) | breech (1, imatinib), pregnancy-induced hypertension (1, imatinib) | none |
| Babu et al. 2015 [30] (article) | India | 3 (3) | imatinib (3) | none | none | none | not reported |
| Breccia et al. 2008 [31] (article) and Pacilli et al. 2009 [51] (conference abstract) | Italy | 5 (5) | imatinib (5) | none | none | podalic position with threatening miscarriage (1, imatinib) | not reported |
| Carlier et al. 2017 [32] (article) and Markarian et al. 2016 [47] (conference abstract) | France | 15 (15)[2] | imatinib (13), nilotinib (1), dasatinib (1) | complex cardiopathy (1, imatinib), hydronephrosis with pyeloureteral junction syndrome (1, imatinib), pulmonary stenosis (1, nilotinib) | spontaneous abortion (1, imatinib), elective termination (2, imatinib, dasatinib) | premature delivery (1, imatinib)[3] | intrauterine growth retardation (1, imatinib), neonatal respiratory distress syndrome (1, imatinib), acute myeloid leukemia (1, nilotinib)[4] |
| Chelysheva et al. 2009 [35], 2011 [34], and 2012 [33] (conference abstracts) | Russia | 14 (14) | imatinib (13), nilotinib (1) | none | none | premature delivery with severe hyperbilirubinemia (1, nilotinib) | not reported |
| Cortes et al. 2008 [36] and 2015 [37] (from the BMS CARES database) (conference abstract and article) | Multinational | 33 (33)[5] | dasatinib (33) | syndactyly (1, dasatinib) | spontaneous abortion (2, dasatinib) | preeclampsia (1, dasatinib)[6] | not reported |
| Cortes et al. 2020 [60] (from the Pfizer safety database) (article) | Multinational | 14 (14)[7] | bosutinib (14) | none | elective termination (4, bosutinib, spontaneous abortion (1)[8] | none | not reported |
| Dou et al. 2019 [59], Jiang et al. 2012 [43] (articles in Chinese)[9] | China | 61 (49) | imatinib (40), nilotinib (5), dasatinib (4) | hypospadiasis (1, imatinib) | elective termination (4, not reported), spontaneous abortion (2, not reported) | premature delivery (1, imatinib) | none |
| Gentile et al. 2014 [39] (article) | Italy | 1 (1) | dasatinib (1) | none | none | placenta accrete (1, dasatinib)[10] | none |
| Grudeva-Popova et al. 2010 [40] (article in Bulgarian) | Bulgaria | 2 (2) | imatinib (2) | none | none | none | none |
| Guerci-Bresler et al. 2011 (from the FI-LMC Group) (article in French) [41] | France | 30 (30) | imatinib (28), nilotinib (2) | none | elective termination (2, imatinib), spontaneous abortion (1, imatinib) | none | not reported |
| Iqbal et al. 2014 [42] (article) | Pakistan | 62 (40) | imatinib (62) | none[11] | elective termination (2, imatinib), stillbirth (1, imatinib)[12] | premature delivery (2, imatinib) | neuroblastoma (2, imatinib)[13] |

*(Continued)*

**Table 2.** (Continued)

| Study population | Country | N⁰ of pregnancies (N⁰ of males) | TKI (N⁰ of pregnancies) | Non-fatal malformations with live births (N⁰ of cases, TKI) | *In utero* fatal events (N⁰ of cases, TKI) | Peripartum feto-maternal complications (N⁰ of cases, TKI) | Infant complications (N⁰ of cases, TKI) |
|---|---|---|---|---|---|---|---|
| Klamova et al. 2013 [44] (conference abstract) | Czech Republic | 9 (8) | imatinib (7), dasatinib (1), unknown (1) | umbilical hernia (1, TKI not reported) | none | none | none (probably) |
| Luciano et al. 2010 [45] (conference abstract) | Italy | 6 (4) | imatinib (6) | none | none | premature delivery (3, imatinib)[14] | none |
| Mukhopadhyay et al. 2015 [49] (article) and Dasgupta et al. 2013 [38] (conference abstract) | India | 4 (4) | imatinib (4) | hydrocephalus (1, imatinib) | elective termination (1, imatinib)[15] | not reported | not reported |
| Madabhavi et al. 2019 [46] (article) and Modi et al. 2018 [48] (conference abstract) | India | 58 (58) | imatinib (58) | none | none | none | none |
| Oweini et al. 2011 [50] (article) | Lebanon | 1 (1) | dasatinib (1) | none | none | not reported | none |
| Ramasamy et al. 2007 [52] (article) | The UK | 5 (4) | imatinib (5) | none | none | none | not reported |
| Ruirui et al. 2016 [53] (article in Chinese) | China | 5 (5) | imatinib (5) | none | spontaneous abortion (1, imatinib) | none | none |
| Shash et al. 2011 [54] (article) | Italy | 2 (1) | imatinib (2) | none | none | none | none |
| Yamina et al. 2015 [56] (conference abstract) | Algeria | 18 (13) | imatinib (15), nilotinib (1), dasatinib (2) | malformation not specified (1, nilotinib) | spontaneous abortion (3, imatinib) | none | not reported |
| Zhou et al. 2013 [57] (article) and Wang et al. 2013 [55] (conference abstract) | China | 7 (7) | imatinib (6), nilotinib (1) | none | none | premature delivery (1, imatinib) | none |
| Xiaohui et al. 2013 [58] (article in Chinese) | China | 1 (1) | imatinib (1) | none | none | not reported | not reported |

TKIs listed were taken at the time of conception or harvesting sperm for cryopreservation.

[1]Follow-up data are available only for patients receiving imatinib.

[2]Diagnosis (chronic myeloid leukemia or gastrointestinal stromal tumor) is not specified in one case.

[3]Twins were delivered at week 36.

[4]The infant who had pulmonary stenosis developed acute leukemia.

[5]Outcomes of 36 cases are unknown.

[6]A healthy baby was delivered at week 37.

[7]Three cases lack data.

[8]Fetal biopsy revealed basal deciduitis with necrotic foci and bleeding.

[9]Four cases discontinued treatment before conception.

[10]A healthy baby was delivered at week 38.

[11]Three cases lack data.

[12]Stillbirth occurred due to fetal malformations.

[13]Twins had a family history positive for neuroblastoma.

[14]The conception in the acute phase treated with imatinib resulted in an uneventful pregnancy and premature delivery.

[15]The conception in the acute phase resulted in an elective abortion. TKI, tyrosine kinase inhibitor.

with live fetus (17 spontaneous abortions, 10 elective terminations, and 1 case of stillbirth) (Fig 2A). Offspring from ten live births (2.5% of total life births) had any malformation (the type of TKI was not specified in one case) (Fig 2B).

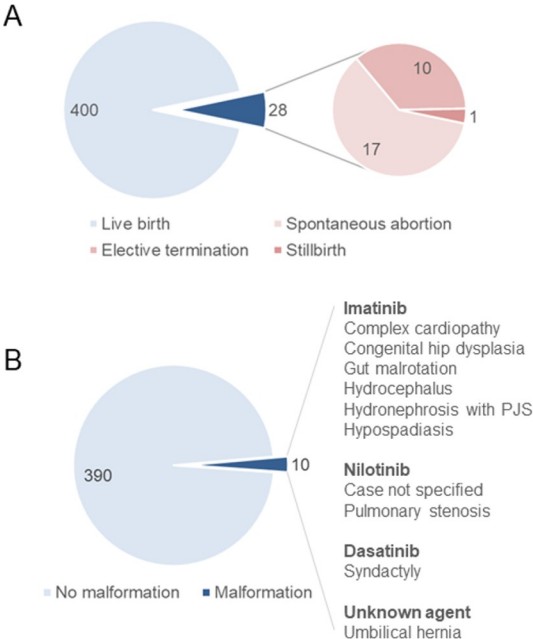

**Fig 2. Characteristics of male patients not discontinuing tyrosine kinase treatment before conception.** A: Distribution of pregnancy outcomes among all pregnancies (n = 428). B: Malformations among live births (n = 400). PJS: pyelouretal junction syndrome.

**Imatinib.**    A total of 327 pregnancies conceived under the effect of IMA, 14 of which (4.3%) did not end up in live birth (six elective and seven spontaneous abortions, one still-birth). The outcomes were not reported by TKI agents separately in one study [59].

Six of 313 live births (1.9%) developed any malformation, these included cases with con-genital hip dysplasia [10, 11, 26], gut malrotation [29], hydronephrosis with pyeloureteral junction syndrome [32, 47], complex cardiopathy [32, 47], hypospadiasis, and hydrocepha-lus [38, 49].

In addition to sporadic cases of breach [27]; pregnancy-induced hypertension [27], podalic position with threatening miscarriage [31, 51], and a total of nine cases of premature delivery were reported. Regarding postnatal complications; cases of jaundice [10, 11, 26], intrauterine growth retardation [32, 47], and neonatal respiratory distress syndrome [32, 47] were reported. In the long-term, one case of neuroblastoma was identified in a child with family history posi-tive for the tumor [42].

**Nilotinib.**    All pregnancies conceived under the effect of NIL ended up in live birth (the outcomes were not reported by TKI agents separately in one study [59]). Two (7.7%) of 26 pregnancies developed malformation; these were a case of pulmonary stenosis [32, 47] and another case in which the malformation was not specified [56].

Regarding feto-maternal complications, one case of premature delivery with severe hyper-bilirubinemia [33–35] and another case of acute myeloid leukemia were reported [32, 47].

**Dasatinib.**    Three out of 46 pregnancies (6.5%) conceived under the effect of DAS ended up in elective termination or spontaneous abortion (the outcomes were not reported by TKI agents separately in one study [59]). One (2.3%) of 43 live births developed syndactyly [37]; otherwise, no malformations were reported.

Regarding feto-maternal complications, one case of preeclampsia [36, 37] and another case of placenta accrete [35] were reported.

**Bosutinib.** Out of 16 pregnancies, four ended up in elective termination and another one in spontaneous abortion, in which basal deciduitis was confirmed [60]. All the other pregnancies were uneventful.

**Ponatinib.** No information is available.

## CML status and conception

Out of 428 pregnancy cases, CML status of 175 fathers were not reported. Among the patients, 250 were in the chronic phase, whereas three patients conceived in the active phase:

- Case 1, aged 31 years, was treated with alternated NIL/IMA in the accelerated phase when conceived (uneventful pregnancy and follow-up) [10, 11, 26].

- Case 2, aged 34 years, was treated with IMA in the blast phase when conceived (uneventful pregnancy, premature delivery at the 34th week) [45].

- Case 3, aged 24 years, treated with IMA 800 mg in the accelerated phase when conceived (elective termination) [38, 49].

Out of the ten malformations, the phase of CML is unknown for five cases, and fathers were in the chronic phase for another five (complete hematological response: two cases; no complete hematological response: one case; and unknown hematological response: two cases). Data on cytogenetic and molecular responses are scarcely reported. However, note that most of the males who had not achieved a complete cytogenetic or molecular response at conception had healthy offspring. Table 3 summarizes data on CML status at conception.

## Quality assessment

The quality of the studies included is summarized in Table 4.

## Discussion

An interesting issue is the safety of exposure to TKIs in men to conceive a pregnancy. Studies suggest it is acceptable to continue TKI with counseling regarding uncertainty, but there are no clear data on safety for men on TKIs to conceive pregnancy (as presented in Table 2). Limited case reports exist of successful, healthy pregnancies conceived by men taking TKI, including second-generation agents (DAS, NIL, and BOS), but there are no reports of successful pregnancies of partners of men on PON. US Food and Drug Administration enrolls TKIs in the 'D' pregnancy category, which means that there is potential evidence of risk on fetal development but, due to the potential benefits of use, the drug may be applied during pregnancy. The labeling does not concern paternal issues, although potentially harmful factors affecting the father and the mother may be associated with fetal development [62].

Congenital anomalies are the leading cause of death in infancy in the US [63]. Based on data from the European Surveillance of Congenital Anomalies (EUROCAT, covering approximately 1.5 million births), major congenital anomalies were reported in 23.9 per 1 000 births (2.39%) between 2003 and 2007; 80% of these cases were live births. Congenital heart defects are the most common anomalies (6.5 per 1 000 births) [64]. In line with these, we observed a total of ten malformations (2.5%), including two heart defects, among the children of those fathers who did not discontinue TKI treatment. Malformations having a little impact on health and function, i.e., the 'minor' anomalies, are included in this number as well (Table 2 and Fig 2) [65]. Taken together, the rate of malformations seems comparable with the European average. However, we must keep in mind that the pattern and incidence of congenital anomalies may vary by region and time, whereas our study population was recruited from many sites

**Table 3. CML status at conception in males with no treatment discontinuation.**

| Study population | N⁰ of pregnancies (N⁰ of males) | TKI (N⁰ of pregnancies) | Disease status at conception | | | |
|---|---|---|---|---|---|---|
| | | | Phase (N⁰ of cases) | Hematological response (N⁰ of cases) | Cytogenetic response (N⁰ of cases) | Molecular response (N⁰ of cases) |
| Abruzzese et al. 2014 [10, 26] (article and conference abstract) and 2016 [11] (article) (from the GINEMA registry) | 44 (probably 40) | imatinib (34), nilotinib (7), dasatinib (1), bosutinib (2) | accelerated (1), chronic (43) | not reported | not reported | not reported |
| Alizadeh et al. 2015 [27] (article) | 10 (5) | imatinib (8), nilotinib (2) | chronic (all) | CHR (all) | CCyR (all) | MMR (all) |
| Aota et al. 2020 [61] (article in Japanese) | 1 (1) | nilotinib | chronic (all) | CHR (all) | CCyR (all) | MMR (all) |
| Assi et al. 2017 [28] (conference abstract) | 7 (7) | nilotinib (5), dasatinib (2) | not reported | not reported | not reported | not reported |
| Ault et al. 2006 [29] (article) | 9 (8) | imatinib (9) | chronic (all) | CHR (all) | not reported | not reported |
| Babu et al. 2015 [30] (article) | 3 (3) | imatinib (3) | not reported | not reported | not reported | not reported |
| Breccia et al. 2008 [31] (article) and Pacilli et al. 2009 [51] (conference abstract) | 5 (5) | imatinib (5) | chronic (all) | CHR (all) | CCyR (4), no CCyR (1) | not reported |
| Carlier et al. 2017 [32] (article) and Markarian et al. 2016 [47] (conference abstract) | 15 (15) | imatinib (13), nilotinib (1), dasatinib (1) | not reported | not reported | not reported | not reported |
| Chelysheva et al. 2009 [35], 2011 [34], and 2012 [33] (conference abstracts) | 14 (14) | imatinib (13), nilotinib (1) | chronic (13), not reported (1) | not reported | not reported | not reported |
| Cortes et al. 2008 [36] and 2015 [37] (from the BMS CARES database) (conference abstract and article) | 33 (33) | dasatinib (33) | not reported | not reported | not reported | not reported |
| Cortes et al. 2020 [60] (from the Pfizer safety database) (article) | 14 (14) | bosutinib (14) | not reported | not reported | not reported | not reported |
| Dou et al. 2019 [59], Jiang et al. 2012 [43] (articles in Chinese) | 61 (49) | imatinib (40), nilotinib (5), dasatinib (4) | chronic (all) | CHR (all) | CCyR (42), no CCyR (7) | MMR (38), no MMR (11) |
| Gentile et al. 2014 [39] (article) | 1 (1) | dasatinib (1) | chronic (all) | CHR (all) | CCyR (all) | MMR (all) |
| Grudeva-Popova et al. 2010 [40] (article in Bulgarian) | 2 (2) | imatinib (2) | chronic (all) | CHR (all) | CCyR (all) | MMR (all) |
| Guerci-Bresler et al. 2011 (from the FI-LMC Group) (article in French) [41] | 30 (30) | imatinib (28), nilotinib (2) | not reported | not reported | not reported | not reported |
| Iqbal et al. 2014 [42] (article) | 62 (40) | imatinib (62) | chronic (all) | CHR (all) | no CyR (6), minor CyR (7), major CyR (17), CCyR (32) | not reported |
| Klamova et al. 2013 [44] (conference abstract) | 9 (8) | imatinib (7), dasatinib (1), unknown (1) | not reported | not reported | not reported | not reported |
| Luciano et al. 2010 [45] (conference abstract) | 6 (4) | imatinib (6) | blast phase (1), chronic (1), not reported (4) | CHR (1), no CHR (1), not reported (4) | CCyR (1), no CCyR (1), not reported (4) | MMR (1), no MMR (1), not reported (4) |
| Mukhopadhyay et al. 2015 [49] (article) and Dasgupta et al. 2013 [38] (conference abstract) | 4 (4) | imatinib (4) | accelerated phase (1), chronic (3) | CHR (2), no CHR (2) | CCyR (1), no CCyR (3) | MMR (0), no MMR (4) |
| Madabhavi et al. 2019 [46] (article) and Modi et al. 2018 [48] (conference abstract) | 58 (58) | imatinib (58) | not reported | not reported | not reported | not reported |
| Oweini et al. 2011 [50] (article) | 1 (1) | dasatinib (1) | not reported | not reported | not reported | not reported |

*(Continued)*

**Table 3.** (Continued)

| Study population | N⁰ of pregnancies (N⁰ of males) | TKI (N⁰ of pregnancies) | Disease status at conception | | | |
|---|---|---|---|---|---|---|
| | | | Phase (N⁰ of cases) | Hematological response (N⁰ of cases) | Cytogenetic response (N⁰ of cases) | Molecular response (N⁰ of cases) |
| Ramasamy et al. 2007 [52] (article) | 5 (4) | imatinib (5) | chronic (all) | CHR (all) | CCyR (2), MCyR (2), PCyR (1) | not reported |
| Ruirui et al. 2016 [53] (article in Chinese) | 5 (5) | imatinib (5) | chronic (all) | CHR (all) | CCyR (4), PCyR (1) | MMR (4), no MMR (1) |
| Shash et al. 2011 [54] (article) | 2 (1) | imatinib (2) | chronic (all) | CHR (all) | CCyR (all) | MMR (all) |
| Yamina et al. 2015 [56] (conference abstract) | 18 (13) | imatinib (15), nilotinib (1), dasatinib (2) | chronic (all) | not reported | not reported | not reported |
| Zhou et al. 2013 [57] (article) and Wang et al. 2013 [55] (conference abstract) | 7 (7) | imatinib (6), nilotinib (1) | chronic (all) | CHR (all) | CCyR (all) | MMR (1), CMR (3), no MR (3) |
| Xiaohui et al. 2013 [58] (article in Chinese) | 1 (1) | imatinib (1) | chronic (all) | CHR (all) | CCyR (all) | MMR (1) |

CCyR, complete cytogenic remission; CHR, complete hematological remission; CMR, complete molecular remission; MCyR, major cytogenic remission; MR, molecular remission; MMR, major molecular remission; PCyR, partial cytogenic remission

worldwide and over a long period, embracing 15 years. The incidence of malformations with IMA (1.9%) is even closer to the European average, but that with NIL is surprisingly high (7.7%). Notably, the latter value must be interpreted with caution due to the low case numbers (Fig 2) and knowing that NIL proved neutral regarding male fertility in rats [12].

The effect of CML status is hard to be assessed since we lacked data in 41% of the cases. If we consider the available data only, five malformations were recorded in patients being in the chronic phase, four of which in those who achieved a complete hematological response. Importantly, several cases without achieving a complete cytogenetic or major molecular response ended up in uneventful pregnancies (Table 3).

## Limitations of the evidence

First, controlled studies are lacking: only case reports and case series studies, which cannot confirm a cause-effect relationship [66], are available. Rechallenge may provide an opportunity to confirm real causality; however, it is not an option in our scenario. Evidence acquired from uncontrolled studies is inherently weak [67]. Hence, we discarded the idea of performing meta-analysis because pooling would not have strengthened the evidence.

Second, reports from large registries with representative populations are lacking, though there are promising initiatives [68, 69].

Third, case reports and case series studies are particularly vulnerable to dissemination bias, questioning the representativeness of the sample. It is impossible to judge whether investigators are more likely to report complicated pregnancies or uncomplicated cases. We tried to reduce publication bias by including non-English language reports [70].

Fourth, the quality of reporting proved to be poor (Table 4)—none of the reports adhered to reporting guidelines [71]. Detailed, long-term follow-up data of the offspring were also lacking (Tables 2 and 4).

Fifth, the recommended sequence of treatment modalities in resistant cases and the first choice of therapy in CML vary with time and across countries. Some cases were pre-treated

**Table 4. Quality assessment.**

| Domain | Leading question | | Comments from the review authors |
|---|---|---|---|
| Selection | Question 1 | Does the patient(s) represent(s) the whole experience of the investigator (center) or is the selection method unclear to the extent that other patients with a similar presentation may not have been reported? | Judged as 'yes' if consecutive patient enrolment was carried out. |
| Ascertainment | Question 2 | Was the exposure adequately ascertained? | Judged as 'yes' if the TKI agent(s), dose(s), and treatment duration were reported. |
| | Question 3 | Was the outcome adequately ascertained? | Judged as 'yes' if the malformation (or its absence) was investigated and described accurately or all pregnancies were uneventful. |
| Casualty | Question 4 | Were other alternative causes that may explain the observation ruled out? | Judged as 'yes' if other potential alternative causes (teratogenic exposure other than TKIs) were ruled out. |
| | Question 5 | Was there a challenge/rechallenge phenomenon? | Not applicable to the review question |
| | Question 6 | Was there a dose-response effect? | Not applicable to the review question |
| | Question 7 | Was follow-up long enough for outcomes to occur? | Judged as 'yes' if at least one-year follow-up of all offspring was reported. Not applicable if offspring were not followed up. |
| Reporting | Question 8 | Is the case(s) described with sufficient details to allow other investigators to replicate the research or to allow practitioners to make inferences related to their own practice? | Judged as 'yes' if the medical history, characteristics, and management of both the fathers and mothers were documented and discussed. |

| Study population | Selection | Ascertainment | | Causality | | | | Reporting |
|---|---|---|---|---|---|---|---|---|
| | Question 1 | Question 2 | Question 3 | Question 4 | Question 5 | Question 6 | Question 7 | Question 8 |
| Abruzzese et al. 2014 [10, 26] (article and conference abstract) and 2016 [11] (article) (from the GINEMA registry) | yes | no | yes | no | N/A | N/A | uncertain | no |
| Alizadeh et al. 2015 [27] (article) | yes | no | yes | no | N/A | N/A | N/A | no |
| Aota et al. 2020 [61] (article in Japanese) | no | yes | yes | no | N/A | N/A | N/A | yes |
| Assi et al. 2017 [28] (conference abstract) | yes | no | yes | no | N/A | N/A | N/A | no |
| Ault et al. 2006 [29] (article) | yes | yes | yes | no | N/A | N/A | yes | yes |
| Babu et al. 2015 [30] (article) | uncertain | yes | yes | no | N/A | N/A | N/A | no |
| Breccia et al. 2008 [31] (article) and Pacilli et al. 2009 [51] (conference abstract) | uncertain | yes | yes | no | N/A | N/A | N/A | no |
| Carlier et al. 2017 [32] (article) and Markarian et al. 2016 [47] (conference abstract) | no | yes | yes | no | N/A | N/A | uncertain | no |
| Chelysheva et al. 2009 [35], 2011 [34], and 2012 [33] (conference abstracts) | uncertain | no | yes | no | N/A | N/A | N/A | no |
| Cortes et al. 2008 [36] and 2015 [37] (from the BMS CARES database) (conference abstract and article) | uncertain | no | no | no | N/A | N/A | N/A | no |
| Cortes et al. 2020 [60] (from the Pfizer safety database) (article) | uncertain | yes | yes | no | N/A | N/A | N/A | yes |
| Dou et al. 2019 [59], Jiang et al. 2012 [43] (articles in Chinese) | yes | yes | yes | no | N/A | N/A | yes | no |
| Gentile et al. 2014 [39] (article) | no | yes | yes | no | N/A | N/A | no | yes |
| Grudeva-Popova et al. 2010 [40] (article in Bulgarian) | no | yes | yes | no | N/A | N/A | yes | yes |
| Guerci-Bresler et al. 2011 (from the FI-LMC Group) (article in French) [41] | uncertain | no | yes | no | N/A | N/A | N/A | no |
| Iqbal et al. 2014 [42] (article) | yes | yes | yes | no | N/A | N/A | uncertain | yes |
| Klamova et al. 2013 [44] (conference abstract) | uncertain | no | yes | no | N/A | N/A | uncertain | no |
| Luciano et al. 2010 [45] (conference abstract) | no | yes | yes | no | N/A | N/A | uncertain | no |
| Mukhopadhyay et al. 2015 [49] (article) and Dasgupta et al. 2013 [38] (conference abstract) | yes | yes | yes | no | N/A | N/A | N/A | yes |
| Madabhavi et al. 2019 [46] (article) and Modi et al. 2018 [48] (conference abstract) | yes | no | yes | no | N/A | N/A | uncertain | no |
| Oweini et al. 2011 [50] (article) | no | yes | yes | no | N/A | N/A | no | yes |
| Ramasamy et al. 2007 [52] (article) | uncertain | yes | yes | no | N/A | N/A | N/A | yes |
| Ruirui et al. 2016 [53] (article in Chinese) | uncertain | yes | yes | no | N/A | N/A | yes | yes |
| Shash et al. 2011 [54] (article) | no | yes | yes | no | N/A | N/A | uncertain | no |
| Yamina et al. 2015 [56] (conference abstract) | yes | yes | no | no | N/A | N/A | N/A | no |
| Zhou et al. 2013 [57] (article) and Wang et al. 2013 [55] (conference abstract) | yes | yes | yes | no | N/A | N/A | uncertain | no |
| Xiaohui et al. 2013 [58] (article in Chinese) | no | yes | yes | no | N/A | N/A | N/A | yes |

N/A; not applicable. TKI; tyrosine-kinase inhibitor.

with non-TKI chemotherapeutics (ancillary therapies, such as interferon) or different TKI agents, raising concerns about stochastic toxicity.

## Implications for clinical practice

The detailed analysis of more than 400 conceptions revealed that the rate of malformations is lower than 3%, comparable with that measured in the general population. However, due to the studies' uncontrolled nature, the level of evidence is low (by the GRADE system). Since discontinuing TKIs may result in CML's progression towards the acute phase while conceiving under the effect of TKIs seems safe, the risk-benefit ratio of TKI discontinuation has not been justified. Semen cryopreservation before TKI treatment may be a much safer alternative. However, due to the weak evidence, we must emphasize the importance of individual risk assessment in daily practice.

## Implications for research

Since none of the studies identified recruited a control group, it is impossible to differentiate the effects of (1) TKIs, (2) ancillary therapies, (3) CML status, and (4) other noxae on pregnancy outcomes. Based only on the frequency of undesirable pregnancy outcomes in our study, we know that statistically strong evidence (powered to the adverse events of TKIs regarding pregnancy) would require a large sample size [72]. CML registries carry the potential to achieve the required size. Controlled observational studies are awaited to verify the safety of TKIs, particularly for the new generation TKIs.

## Author Contributions

**Conceptualization:** Zsolt Szakács, Péter Jenő Hegyi, Alizadeh Hussain.

**Data curation:** Zsolt Szakács, Péter Jenő Hegyi, Márta Balaskó, Adrienn Erős, Szabina Szujó, Judit Pammer, Bernadett Mosdósi, Mária Simon, Gabriella Für.

**Formal analysis:** Nelli Farkas.

**Funding acquisition:** Péter Hegyi.

**Investigation:** Judit Pammer, Mária Simon, Arnold Nagy, Alizadeh Hussain.

**Methodology:** Zsolt Szakács, Judit Pammer, Bernadett Mosdósi, Arnold Nagy.

**Supervision:** Péter Hegyi, Bernadett Mosdósi, Alizadeh Hussain.

**Validation:** Péter Hegyi.

**Visualization:** Nelli Farkas.

**Writing – original draft:** Zsolt Szakács, Péter Hegyi, Márta Balaskó, Mária Simon, Alizadeh Hussain.

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
