## [Decision Letter · Decision Letter 0]

15 Oct 2020

PONE-D-19-31926

Pregnancy outcomes of males with chronic myeloid leukemia treated with tyrosine-kinase inhibitor therapy: A systematic review

PLOS ONE

Dear Dr. Szakács,

Thank you for submitting your manuscript to PLOS ONE. After careful consideration, we feel that it has merit but does not fully meet PLOS ONE’s publication criteria as it currently stands. Therefore, we invite you to submit a revised version of the manuscript that addresses the points raised during the review process.

Please submit your revised manuscript within 30 days. If you will need more time than this to complete your revisions, please reply to this message or contact the journal office at plosone@plos.org. Please include the following items when submitting your revised manuscript:

We look forward to receiving your revised manuscript.

Kind regards,

Ashkan Emadi, MD, PhD

Academic Editor

PLOS ONE

Journal Requirements:

2. We noticed you have some minor occurrence(s) of overlapping text with the following previous publication(s), which needs to be addressed:

https://doi.org/10.1097/01.COT.0000535069.50384.64

https://doi.org/10.1177%2F1078155217710553

In your revision ensure you cite all your sources (including your own works), and quote or rephrase any duplicated text outside the Methods section. Further consideration is dependent on these concerns being addressed.

Additional Editor Comments (if provided):

Dear Dr. Szakács,

Your typescript was reviewed by the experts in the field. Your article can be considered for publication in PLOS ONE after minor revision. Please see the Reviewers' comments.

Sincerely,

Ashkan Emadi, MD, PhD

Reviewers' comments:

Reviewer's Responses to Questions

**Comments to the Author**

1. Is the manuscript technically sound, and do the data support the conclusions?

Reviewer #1: Yes

Reviewer #2: Yes

2. Has the statistical analysis been performed appropriately and rigorously? 

Reviewer #1: N/A

Reviewer #2: Yes

3. Have the authors made all data underlying the findings in their manuscript fully available?

Reviewer #1: Yes

Reviewer #2: Yes

4. Is the manuscript presented in an intelligible fashion and written in standard English?

Reviewer #1: Yes

Reviewer #2: Yes

5. Review Comments to the Author

Reviewer #1: In this manuscript Szakacs and colleagues have conducted a systematic review of literature assessing the association between TKI therapy and pregnancy outcomes in CML male patients.

The authors performed diligent systematic research using 7 electronic databases and found 35 potentially eligible manuscripts. Final 25 manuscripts were included in the review (after removal of over lapping study populations.

Of the 362 pregnancies studies, 320 pregnancies were fathered w/o TKI discontinuation while 345 ended in live births. 10 pregnancies (2.9%) ended up in fetal malformations. This is comparable with the rate of malformations in the general population. Moreover the rate of malformations noted were highest with nilotinib. I agree with the authors that this info should be used with caution since only 20 live births were on patients taking nilotinib. The authors also found no impact of continuing TKI therapy at the time of conception and thereafter on the rate of malformation.

This is an overall well written manuscript. I recommend it to be accepted.

Reviewer #2: Overall, this is a well-conducted systematic review on an important topic that often seems overlooked. A few comments that I think may strengthen the manuscript:

1) I think the title can be revised - "Pregnancy outcomes of males..." sounds as though the man is carrying the child. Perhaps something along the lines of "Males who father children on TKIs" or "offspring of men treated with TKIs"

2) Similarly, I think there are several areas in the manuscript where the English is imprecise, or sentences are incomplete (fragments) - for example, page 21, line 300 "Nor there was the analysis of dose-response" is a fragment, as is page 22 line 319: "Not to mention the lack of detailed, long-term follow-up data on the offspring..."

3) Table 2: In the table it seems that the numbers skip from 1 (Abruzzese, under "infant complications") to 4 (Carlier, under "No of pregnancies").

4) If available, on page 12 under "Planned treatment discontinuation", some information on low long TKI was discontinued would be helpful to have. Did these patients stop TKI 1 week before trying to conceive? 6 months?

5) For figure 2, the various shades of blue are VERY hard to distinguish. In addition, I think fig 2B and 2D should be omitted - the information is available in the text and is not difficult to comprehend, so I do not think these two figures are much of a visual aid.

6) Reasons for elective termination of pregnancies should be added, if available. Although numbers are obviously limited, this information may shed some light on other consequences of TKI treatment and fathering children.

7) Regarding the discussion, page 20, 2nd full paragraph (lines 272-286) is somewhat repetitive as this information is mostly covered in the introduction. If the authors elect to keep this paragraph, it seems out of place - it would probably fit better earlier in the discussion, maybe as the first paragraph.

8) The first several sentences of the 3rd full paragraph on page 20, starting with "If a couple desires fertility..." is not based on the data that the authors present in this manuscript, nor on data referenced in the literature. If recommendations are to be made based on the authors' opinions, this should be in the "implications for clinical practice" paragraph, and should clearly state that these are based on the authors' experience/opinion.

9) Lastly, the section on "limitations of the evidence" in the discussion is much too long and the authors could afford to be more concise here.

6. PLOS authors have the option to publish the peer review history of their article (what does this mean?). If published, this will include your full peer review and any attached files.

Reviewer #1: **Yes: **Kiran Naqvi

Reviewer #2: No

---

## [Author Response · Author response to Decision Letter 0]

3 Nov 2020

Rebuttal letter

Reviewer #1: In this manuscript Szakacs and colleagues have conducted a systematic review of literature assessing the association between TKI therapy and pregnancy outcomes in CML male patients.

The authors performed diligent systematic research using 7 electronic databases and found 35 potentially eligible manuscripts. Final 25 manuscripts were included in the review (after removal of over lapping study populations.

Of the 362 pregnancies studies, 320 pregnancies were fathered w/o TKI discontinuation while 345 ended in live births. 10 pregnancies (2.9%) ended up in fetal malformations. This is comparable with the rate of malformations in the general population. Moreover the rate of malformations noted were highest with nilotinib. I agree with the authors that this info should be used with caution since only 20 live births were on patients taking nilotinib. The authors also found no impact of continuing TKI therapy at the time of conception and thereafter on the rate of malformation.

This is an overall well written manuscript. I recommend it to be accepted.

[Authors’ comment]: We appreciate the kind words. Thank you for the time spent reviewing our paper.

 

Reviewer #2: Overall, this is a well-conducted systematic review on an important topic that often seems overlooked. A few comments that I think may strengthen the manuscript:

[Authors’ comment]: We greatly appreciated the comments, which, we believe, have significantly improved the manuscript.

1) I think the title can be revised - “Pregnancy outcomes of males...” sounds as though the man is carrying the child. Perhaps something along the lines of “Males who father children on TKIs” or “offspring of men treated with TKIs”

[Authors’ reply]: Indeed, our title may be misunderstood. We rephrased it to ’Pregnancy outcomes of women whom spouse fathered children after tyrosine-kinase inhibitor therapy for chronic myeloid leukemia: A systematic review’.

2) Similarly, I think there are several areas in the manuscript where the English is imprecise, or sentences are incomplete (fragments) - for example, page 21, line 300 “Nor there was the analysis of dose-response” is a fragment, as is page 22 line 319: “Not to mention the lack of detailed, long-term follow-up data on the offspring...”

[Authors’ reply]: We revised the manuscript and attempted to improve the English language.

3) Table 2: In the table it seems that the numbers skip from 1 (Abruzzese, under “infant complications”) to 4 (Carlier, under “No of pregnancies”).

[Authors’ reply]: We apologize for the oversight. Now in Table 2, superscript numbers read in ascending order from left to right and top to bottom. Besides, footnotes and abbreviations were simplified to ease understanding.

4) If available, on page 12 under “Planned treatment discontinuation”, some information on low long TKI was discontinued would be helpful to have. Did these patients stop TKI 1 week before trying to conceive? 6 months?

[Authors’ reply]: Thank you for the idea. We added a column to Table 1, where we report these data. The time intervals ranged between 15 days to 5 months.

5) For figure 2, the various shades of blue are VERY hard to distinguish. In addition, I think fig 2B and 2D should be omitted - the information is available in the text and is not difficult to comprehend, so I do not think these two figures are much of a visual aid.

[Authors’ reply]: We removed Figs 2B and 2D, which resolved the issue with the hard recognition of the various shades of blue.

6) Reasons for elective termination of pregnancies should be added, if available. Although numbers are obviously limited, this information may shed some light on other consequences of TKI treatment and fathering children.

[Authors’ reply]: We revised the papers for the information. Unfortunately, only Mukhopadhyay et al. 2015) reported the cause of elective termination in the case of an unplanned pregnancy.

7) Regarding the discussion, page 20, 2nd full paragraph (lines 272-286) is somewhat repetitive as this information is mostly covered in the introduction. If the authors elect to keep this paragraph, it seems out of place - it would probably fit better earlier in the discussion, maybe as the first paragraph.

[Authors’ reply]: We agree on the point so that almost the entire content of the paragraph mentioned has been removed.

8) The first several sentences of the 3rd full paragraph on page 20, starting with “If a couple desires fertility...” is not based on the data that the authors present in this manuscript, nor on data referenced in the literature. If recommendations are to be made based on the authors’ opinions, this should be in the “implications for clinical practice” paragraph, and should clearly state that these are based on the authors’ experience/opinion.

[Authors’ reply]: Fair point. The paragraph mentioned has been removed. The concept of semen cryopreservation before TKI treatment, as an alternative of planned treatment discontinuation, is described in the “implications for clinical practice” paragraph.

9) Lastly, the section on “limitations of the evidence” in the discussion is much too long and the authors could afford to be more concise here.

[Authors’ reply]: The section mentioned was shortened from 393 to 208 words.

---

## [Editor Report · Decision Letter 1]

16 Nov 2020

Pregnancy outcomes of women whom spouse fathered children after tyrosine kinase inhibitor therapy for chronic myeloid leukemia: A systematic review

PONE-D-19-31926R1

Dear Dr. Szakács,

We’re pleased to inform you that your manuscript has been judged scientifically suitable for publication and will be formally accepted for publication once it meets all outstanding technical requirements.

Kind regards,

Ashkan Emadi, MD, PhD

Academic Editor

PLOS ONE

---

## [Editor Report · Acceptance letter]

18 Nov 2020

PONE-D-19-31926R1 

Pregnancy outcomes of women whom spouse fathered children after tyrosine kinase inhibitor therapy for chronic myeloid leukemia: A systematic review 

Dear Dr. Szakács:

I'm pleased to inform you that your manuscript has been deemed suitable for publication in PLOS ONE. Congratulations! Your manuscript is now with our production department. 

Kind regards, 

on behalf of

Dr. Ashkan Emadi 

Academic Editor

PLOS ONE